# Dysregulated Neutrophil Phenotype and Function in Hospitalised Non-ICU COVID-19 Pneumonia

**DOI:** 10.3390/cells11182901

**Published:** 2022-09-16

**Authors:** Kylie B. R. Belchamber, Onn S. Thein, Jon Hazeldine, Frances S. Grudzinska, Aduragbemi A. Faniyi, Michael J. Hughes, Alice E. Jasper, Kay Por Yip, Louise E. Crowley, Sebastian T. Lugg, Elizabeth Sapey, Dhruv Parekh, David R. Thickett, Aaron Scott

**Affiliations:** 1Birmingham Acute Care Research Group, Institute of Inflammation and Ageing, University of Birmingham, Birmingham B15 2TT, UK; 2National Institute for Health Research Surgical Reconstruction and Microbiology Research Centre, Queen Elizabeth Hospital Birmingham, Birmingham B15 2TH, UK; 3PIONEER HDR-UK Hub in Acute Care, Institute of Inflammation and Ageing, University of Birmingham, Birmingham B15 2TT, UK; 4NIHR Clinical Research Facility, University Hospitals Birmingham NHS Foundation Trust, Edgbaston, Birmingham B12 2GW, UK

**Keywords:** inflammation, COVID-19, neutrophil, innate immunity

## Abstract

**Rationale**: Infection with the SARS-CoV2 virus is associated with elevated neutrophil counts. Evidence of neutrophil dysfunction in COVID-19 is based on transcriptomics or single functional assays. Cell functions are interwoven pathways, and understanding the effect across the spectrum of neutrophil function may identify therapeutic targets. **Objectives**: Examine neutrophil phenotype and function in 41 hospitalised, non-ICU COVID-19 patients versus 23 age-matched controls (AMC) and 26 community acquired pneumonia patients (CAP). **Methods**: Isolated neutrophils underwent ex vivo analyses for migration, bacterial phagocytosis, ROS generation, NETosis and receptor expression. Circulating DNAse 1 activity, levels of cfDNA, MPO, VEGF, IL-6 and sTNFRI were measured and correlated to clinical outcome. Serial sampling on day three to five post hospitalization were also measured. The effect of ex vivo PI3K inhibition was measured in a further cohort of 18 COVID-19 patients. **Results**: Compared to AMC and CAP, COVID-19 neutrophils demonstrated elevated transmigration (*p* = 0.0397) and NETosis (*p* = 0.0332), and impaired phagocytosis (*p* = 0.0036) associated with impaired ROS generation (*p* < 0.0001). The percentage of CD54+ neutrophils (*p* < 0.001) was significantly increased, while surface expression of CD11b (*p* = 0.0014) and PD-L1 (*p* = 0.006) were significantly decreased in COVID-19. COVID-19 and CAP patients showed increased systemic markers of NETosis including increased cfDNA (*p* = 0.0396) and impaired DNAse activity (*p* < 0.0001). The ex vivo inhibition of PI3K γ and δ reduced NET release by COVID-19 neutrophils (*p* = 0.0129). **Conclusions**: COVID-19 is associated with neutrophil dysfunction across all main effector functions, with altered phenotype, elevated migration and NETosis, and impaired antimicrobial responses. These changes highlight that targeting neutrophil function may help modulate COVID-19 severity.

## 1. Introduction

Coronavirus disease 2019 (COVID-19), caused by the severe acute respiratory syndrome coronavirus 2 (SARS-CoV2) virus, was declared a global pandemic by The World Health Organization (WHO) on 11 March 2020 [1]. Up to 80% of people infected with SARS-CoV2 experience mild to moderate respiratory disease, but in 10–20% of cases, infection can manifest as pneumonitis, with 5% progressing to acute respiratory distress syndrome (ARDS). The overall worldwide mortality rate is 2.2%, but this increases with the development of pneumonia and acute respiratory distress syndrome (ARDS)of ARDS [2].

Dysregulated virus induced host-immune responses are thought to be the primary cause of severe COVID-19 [3]. Neutrophils are frontline effector cells that protect against rapidly dividing pathogens and play a pivotal role in the antiviral immune response [4]. In the early stages of an inflammatory event, neutrophils migrate into lung tissue where they perform phagocytosis and release proteases, reactive oxygen species (ROS), and, as a later response, neutrophil extracellular traps (NETs) to aid in the clearance of infection [5].

Advanced age is a recognised risk factor for severe COVID-19, including the development of ARDS [6,7,8]. Ageing is associated with changes in neutrophil function, including reduced migratory accuracy [9], phagocytosis [10] and NETosis [11], which may delay pathogen clearance and increase bystander tissue injury. Further impairments in neutrophil function have been observed in patients with pneumonia [12], sepsis [13] and ARDS [14]. This dysfunction is most apparent in older adults with severe infections [15] and is associated with poor clinical outcomes [12,13]. Importantly, these functional deficits appear amenable to therapeutic correction, especially in early or less severe infective episodes, suggesting that neutrophils, or their products, may form a tractable target in inflammatory disease [16,17]. There is some evidence that neutrophil dysfunction could be associated with distinct cellular phenotypes within the neutrophil population, with immature, senescent, activated and pro-inflammatory neutrophils identified in age and disease [13,18,19].

Emerging studies suggest that neutrophils are implicated in the pathogenesis of severe COVID-19 and every reporting study has described cell dysfunction (or inferred cell dysfunction through transcriptomics), which could contribute to tissue damage and secondary infection [20,21,22,23,24,25,26,27] However, studies have often been small, have failed to provide age-matched controls, assessed only patients on the intensive care unit (ICU) after a considerable delay since symptom onset, or considered neutrophil functions in isolation [28].

These are important considerations. As neutrophil functions change with age, age matched controls are important in identifying pathological differences. Although approximately 12% of hospitalized COVID-19 patients require ICU support, the majority of hospital admissions and deaths occur on non-ICU wards [29]; understanding how to improve outcomes for this patient group is vital. Most studies in sepsis have demonstrated that cell function is more therapeutically tractable during early disease (before ICU care is needed), so the timeliness of intervention is important. Cell functions are enabled by interwoven cell pathways with important differences in internal signalling; knowing which functions are impaired informs which therapeutic strategies might restore/maintain all facets of cellular function.

We hypothesised that neutrophils from COVID-19 patients would exhibit diverse altered effector functions and changes to phenotype, with the degree of dysfunction associated with adverse patient outcomes.

This study aimed to perform, for the first time, a comprehensive assessment of ex vivo neutrophil phenotypes and functions in a statistically powered cohort of hospitalized, non-ICU SARS-CoV2 infected patients compared to aged-matched controls and patients with non-COVID community acquired pneumoniae (CAP) and to investigate relationships with clinical outcomes.

## 2. Methods

### 2.1. Healthy Donor and Patient Recruitment

Recruitment is summarized in Figure 1. COVID-19 patients were recruited from January to March 2021, while community-acquired pneumonia (CAP) patients were recruited from August 2021–January 2022 from the Queen Elizabeth Hospital, Birmingham, in accordance with ethics REC ref: 19/WA/0299 and 20/WA/0092 approved by the West Midlands—Solihull research ethics committee. A further cohort of 18 COVID-19 patients was recruited for inhibitor studies between August 2021–January 2022. Written informed consent was obtained where possible; patients unable to consent due to lack of capacity were either consented by proxy, designated consultee via telephone or professional consultee. Follow-up samples were collected at days three to five with confirmed consent where possible (Appendix A).

COVID-19 patients were recruited within 48 h of hospital admission due to pneumonitis/pneumonia related to COVID-19. All patients had a positive COVID-19 PCR swab. No patients received novel treatments or were part of a COVID clinical medicinal trial on recruitment to this study. Exclusion criteria is listed in Appendix A. COVID-19 patients were stratified using the 4C Mortality Score for COVID-19, separated by scores <9 and ≥9 (in-hospital mortality <9.9% and >31.4% respectively) [30]. Patients were classified with ARDS based on the Berlin criteria; SpO_2_: FiO_2_ ratio (SF), converted to PaO_2_: FiO_2_ (PF) (SF = 57-0.61PF).

CAP patients were recruited within 48 h of hospital admission due to non-COVID-19 pneumonia. All patients had a negative COVID-19 swab. Exclusion criteria is listed in the Appendix A. CAP patients were stratified using the CURB-65 score, separated by scores ≤2 and 3+ (in hospital mortality 6.8% vs. 14% respectively).

Age matched controls (AMC) were either recruited from patients attending pre-booked face-to-face outpatient appointments or from hospital staff. AMC had no evidence of acute illness, including COVID-19, within the last two weeks, as assessed by a respiratory physician, and met the other exclusion criteria.

### 2.2. Neutrophil Phenotypic and Functional Analysis

Peripheral blood samples were taken, and plasma and serums were stored [13]. Neutrophils were isolated from peripheral blood as previously described [9]. Neutrophils underwent phenotypical analysis by flow cytometry (Appendix A), and functional analysis, which included transwell migration, phagocytosis of fluorescent *S. pneumoniae* and NETosis by the release of cell-free (cf) DNA (Appendix A). Plasma samples underwent analysis for cfDNA quantification, citrullinated histone H3 detection and biomarker quantification, whilst DNase activity was measured in serum. Full methods are found in Appendix A. Figure legends represent the number of samples per experiment, with experiments performed according to the number of isolated neutrophils per patient. All functional data relates to COVID-19 cohort 1, with inhibitor data relating to cohort 2.

### 2.3. Statistics

A statistical analysis was performed using Prism v9.0.0 (GraphPad Software Inc., San Diego, CA, USA). A Kolmogorov-Smirnov Test was used to determine data distribution. Normally distributed data were analysed using a student’s *t*-test or ANOVA. A Mann-Whitney U test for unpaired data, a Wilcoxon test for paired data, or a Kruskall-Wallis test was used to analyse non-normally distributed data. Data are presented throughout as median (IQR), with each n number representing a separate study participant. Significance was defined at *p* < 0.05. There were no corrections for multiple comparisons, but exact *p* values are given. A power calculation performed on isolated neutrophil NETosis data (80%, alpha 0.05) suggested that 18 participants were required in each group (see Appendix A for details).

## 3. Results

### 3.1. Clinical Characterisation

41 COVID-19 patients (mean age 71.5 years), two healthy AMC (mean age 70 years) and 26 CAP patients (mean age 67.5 years) were included in the study. Demographics are provided in Table 1. COVID-19 patients were admitted to hospital seven days (range 3–14) after symptom onset and were recruited to the study after a median of two days (range 1–2). Length of hospital stay was 7.7 days (survivors 7.5 days, non-survivors 9.1 days), and the mortality rate was 24% (10/41). 38/41 patients received dexamethasone as part of their acute treatment as per standard of care, but none received other novel COVID-19 treatments [31]. 17/41 patients had ARDS as defined by the Berlin criteria with the exception of ventilation pressure [32], and of these, eight had moderate to severe ARDS. 3/41 patients were transferred to ICU after recruitment. For inhibitor studies, a further cohort of 18 COVID-19 patients (mean age 71 years) were included in the study. Demographics are provided in Table 1. For analysis, demographics of COVID-19 and CAP patients separated by 4C score or curb65 score (Table 1) or survival (Appendix A) was performed.

### 3.2. Neutrophil Migration Is Elevated through a Transwell System in COVID-19

Neutrophils from COVID-19 patients demonstrated increased transwell migration towards CXCL-8 compared to AMC (fold change of neutrophils migrated to CXCL-8 vs. vehicle control: 12.15 (51.4) AMC vs. 40.63 (115.2) COVID-19, *p* = 0.0332) and CAP (9.55 (23.7, *p* = 0.0332, Figure 2A). Migration was not associated with 4C score (*p* = 0.5575, Appendix A), or survival at 28 days (*p* = 0.2563, Appendix A). There was no change in migration at follow up (*p* = 0.1641, Appendix A).

### 3.3. Neutrophil Phagocytosis Is Impaired in COVID-19

Neutrophil phagocytosis of *S. pneumoniae* was significantly decreased in COVID-19 patients compared to AMC (Median fluorescence intensity (MFI): 8.0 (4.2) AMC vs. 6.6 (2.6) COVID-19, *p* = 0.0366) and CAP (MFI 9.8 (4.8), *p* = 0.0052, Figure 2B). Phagocytosis was not associated with 4C score (*p* = 0.3257, Appendix A) or survival at 28 days (*p* = 0.9228, Appendix A). There were no differences in neutrophil viability between patient groups or treatment conditions (*p* = 0.6726, Appendix A).

### 3.4. Neutrophil Derived ROS Generation following Phagocytosis Is Impaired in COVID-19

Cytoplasmic (c)ROS, and nuclear/mitochondrial (n/m)ROS were measured in resting neutrophils and following phagocytosis. Compared to unstimulated cells, cROS levels were elevated after phagocytosis in both AMC (MFI: 46.8 (28) baseline vs. 64.9 (51) 30 min, *p* = 0.0091), and CAP patients (MFI: 285 (211) baseline vs. 357 (260) 30 min, *p* = 0.038), but not in COVID-19 patients (MFI: 44.1 (35) baseline vs. 65.3 (60) 30 min, *p* = 0.134, Figure 2C). CAP patients had significantly higher cROS at baseline, and after 30 min phagocytosis compared to both AMC and COVID-19 patients (*p* < 0.0001, Figure 2C).

Compared to resting neutrophils, n/mROS levels were significantly higher after phagocytosis in neutrophils isolated from AMC (MFI: 21.8 (21) baseline vs. 32.0 (38), *p* < 0.0001), but not in COVID-19 patients (MFI: 18.9 (12) baseline vs. 21.2 (12), *p* = 0.0329) or CAP patients (MFI: 22.2 (21) baseline vs. 26.7 (17), *p* = 0.989, Figure 2D). No significant differences were found in the levels of n/mROS in resting neutrophils; however, compared to AMC, both COVID-19 neutrophils (*p* < 0.0001) and CAP neutrophils (*p* = 0.0101, Figure 2D) displayed reduced levels of n/mROS after phagocytosis.

### 3.5. Neutrophil NETosis Is Elevated in COVID-19

A larger increase in the level of cfDNA was detected in supernatants obtained from COVID-19 patient neutrophils post-PMA treatment compared to AMC (fold change in absorbance of neutrophils stimulated with PMA vs. vehicle control: 1.29 (0.32) AMC vs. 1.53 (1.66) COVID-19, *p* = 0.0394, Figure 3A). There were no differences in resting neutrophils NET production between COVID patients, CAP patients and AMC (Appendix A). Severe disease was associated with increased NETosis (fold change: 1.17 (0.35) Low 4C vs. 1.41 (0.80) High 4C, *p* = 0.0118, Figure 3B).

At hospital admission, COVID-19 patients presented with higher concentrations of plasma cfDNA compared to AMC (621 ng/mL (1324) AMC vs. 1071 ng/mL (856) COVID-19, *p* = 0.0396, Figure 3C), which persisted at day three to five (*p* = 0.0186, Figure 3D). CAP patients also had higher levels of plasma cfDNA compared to AMC (1220 ng/mL (686), *p* = 0.0322).

To determine whether neutrophils, via NETosis, were a source of the cfDNA, plasma samples were screened for the presence of CitH3, a protein that decorates the DNA backbone of NETs [33]. Western blotting revealed the presence of CitH3 in six out of eight samples analyzed (75%, Appendix A).

Using NETs as a substrate, serum DNase activity was lower in COVID-19 patients at the time of hospital admission when compared to both AMCs and CAP patients (% degradation: 88.4% (93) AMC vs. 12.8% (49) COVID-19, *p* < 0.0001, vs. 77.45% (34) CAP, *p* < 0.0001, Figure 3E).

### 3.6. Neutrophil Phenotype Is Altered in COVID-19

To determine whether changes observed in neutrophil function were associated with phenotype, expression of key surface molecules were investigated by flow cytometry. This was compared to both AMC and CAP patients. A table of percentage receptor expression and MFI is shown in Table 2.

Compared to AMC, both the percentage of neutrophils expressing the activation marker CD11b (82% (15) AMC vs. 68% (44) COVID-19, *p* = 0.0014, Figure 4A), and its surface expression (MFI: 399 (331) AMC vs. 40 (223) COVID-19, *p* = 0.0026, Figure 4B) were reduced in COVID-19 patients. Compared to CAP patients, the expression of CD11b was also reduced in COVID-19 (% expression 83.9% (24) CAP, *p* = 0.046; MFI:1701 (1268) CAP, *p* < 0.0001). 

The percentage of neutrophils expressing CD54, a marker of reverse transmigration, was elevated in COVID-19 patients compared to both AMC and CAP patients (26% (37) AMC vs. 71% (21) COVID-19, *p* < 0.0001, vs. 7% (11) CAP, *p* < 0.0001, Figure 4C).

The percentage of neutrophils expressing CD66b, CD62L, CD10, CXCR2, CXCR4, CD11c and PD-L1 did not differ between AMC and COVID-19 patients (see Table 2). The surface expression of CD66b, CD62L, CD10, CXCR2, CXCR4 and CD11c did not differ between AMC and COVID-19 patients (see Table 2). The surface expression of PD-L1 was significantly reduced in COVID-19 patients compared to AMC and CAP (MFI 193 (210) AMC vs. 70 (132), COVID-19, *p* < 0.006, vs. 524 (264) and CAP, *p* < 0.0001, Figure 4D).

There was no association of neutrophil phenotypic marker expression with 4C severity score, or in survivors and non-survivors.

On day three to five follow-up, relative to baseline readings, there was a decrease in CXCR2 expression: (MFI: 1960 (1601) day one vs. 1126 (1262), day three to five, *p* = 0.0322, Figure 4E) and a significant increase in CXCR4 expression: (MFI: 1785 (3951) day one vs. 10,248 (16,483), day three to five, *p* = 0.0273, Figure 4F). The percentage of cells expressing CD54 was significantly reduced at three to five day follow up: (80.6% (34.6) day 1 vs. 58.9% (38.3), Day three to five, *p* = 0.0420, Figure 4G).

On day three to five follow-up, relative to baseline readings, there was a decrease in CXCR2 expression: (MFI: 1960 (1601) day 1 vs. 1126 (1262), day three to five, *p* = 0.0322, Figure 4E) and a significant increase in CXCR4 expression: (MFI: 1785 (3951) day one vs. 10,248 (16,483), day three to five, *p* = 0.0273, Figure 4F). The percentage of cells expressing CD54 was significantly reduced at three to five day follow up: (80.6% (34.6) Day one vs. 58.9% (38.3), day three to five, *p* = 0.0420, Figure 4G).

### 3.7. Systemic Inflammatory Mediators Are Elevated in COVID-19

Compared to AMC, both COVID-19 and CAP patients showed elevated levels of IL-6 (COVID-19 *p* = 0.0296, CAP *p* = 0.0106, Figure 5A), VEGF (COVID-19 *p* < 0.0001, CAP *p* = 0.0032, Figure 5B), and sTNFRI (COVID-19 *p* < 0.0001, CAP *p* < 0.0001, Figure 5C). Compared to AMC, only COVID-19 patients showed elevated MPO (*p* < 0.0001, Figure 5D). No difference in levels of GM-CSF (Figure 5E) were observed.

Only 2/41 COVID-19 patients demonstrated a hyperinflammatory phenotype when stratified using the algorithm developed to classify non-COVID ARDS phenotypes [34]. IL-6 and sTNFRI concentrations were raised in those patients with a 4C score ≥ 9 compared to a score <8 (IL-6 *p* = 0.0059, sTNFRI *p* = 0.0478, Appendix A). IL-6 levels were significantly raised in patients with moderate to severe ARDS compared to mild (*p* = 0.0468, Appendix A).

### 3.8. Pi3K Inhibitors

Pre-incubation of COVID-19 neutrophils from cohort 2, with Pi3k delta inhibitor CAL101 and gamma inhibitor AS252434 significantly decreased PMA induced NETosis (PMA RFU 247 (228.5) vs. delta 207.5 (242.5), *p* = 0.0129, vs. gamma 216.5 (242), *p* = 0.0156, Figure 6). There was no significant effect of inhibitors on phagocytosis or transwell migration (Appendix A).

## 4. Discussion

We present novel findings of COVID-associated neutrophil dysfunction across all main effector functions. In summary, compared to AMC and patients with CAP, systemic neutrophils from patients hospitalized with moderate severity COVID-19 demonstrated increased migration, impaired anti-microbial responses including reduced phagocytosis and nuclear/mitochondrial ROS generation after phagocytosis. Later/end phase neutrophil responses were increased, namely ex vivo NETosis with evidence of increased systemic NETosis, coupled with reduced DNase activity, which was also elevated in CAP. We also show an altered but distinct neutrophil phenotype, not compatible with a purely activated, immature, senescent or anti-inflammatory phenotype as described before (results summarised in Figure 7). Our data suggests the energetics of the cells were not overtly compromised, as some “high energy-consuming” functions (such as migration [35]) were elevated. Of note, some of these changes have been described by authors studying COVID-19 before the widespread use of dexamethasone as standard of care [27,36], suggesting that our results are not a treatment effect.

Individually, as described in other studies, these changes to effector function could compromise aspects of the host defence. Collectively, these changes represent a clear mechanism for significant tissue damage. Poor phagocytosis would impede pathogen clearance, increasing the likelihood of secondary infection and amplifying inflammation. NETosis is implicated in tissue damage and thrombotic events in several disease settings [37,38]. The inability to clear NETs through reduced DNAse activity would further augment NETosis-associated tissue damage [39].

Secondary infection in COVID-19 is associated with increased severity of lung disease and poorer outcomes [40,41]. Impaired neutrophil antimicrobial responses towards *S. pneumoniae*, the most common bacteria implicated in secondary infection in COVID-19 [42], alongside impaired intracellular ROS generation, which is important for phagosomal bacterial killing [43], may contribute to the incidence of secondary infection and poorer outcomes for these patients.

Elevated NETosis [23,24,44] and increased systemic concentrations of cfDNA [23,24] have been described previously in small numbers of COVID-19 patients and in CAP [45]. Our observation of reduced serum DNase activity confirms a previous study [46] and builds on studies showing reduced plasma concentrations of Gelsolin in COVID-19; Gelsolin depolymerizes filamentous actin, an inhibitor of DNAse activity [47,48,49]. Thus, a circulating microenvironment dominated by negative regulators of DNase-1 could offer a potential mechanistic explanation for the impaired DNase-1 activity we report, with elevated NETosis contributing to host tissue damage and thrombotic events. Indeed, disulfiram, a drug that blocks gasdermin D (important for NET formation), reduced NET production and neutrophil infiltration to the lungs of SARS-CoV2 infected hamsters, suggesting that such therapies may be beneficial in the treatment of COVID-19 [50].

The collective pattern of neutrophil dysfunction in COVID-19 speaks of alterations to mechanosensing within these cells. Elevated migration and impaired phagocytosis could both be linked to reduced pseudopod extrusion, which is known to increase migratory speed [51]. Furthermore, pseudopods are involved in phagocytosis, with reorganization of the cell composition to enable bacterial engulfment [52]. Phosphoinositide 3-kinase (PI3K) is a key intracellular signalling molecule involved in chemotaxis, cytoskeletal rearrangement for phagocytosis, and superoxide generation [53], and has most recently been implicated in NETosis [54]. Aberrant PI3K signalling is linked to increasing age, and PI3K inhibitors have been shown to improve neutrophil migratory accuracy in the elderly [9] while reducing NET formation ex vivo [55,56]. Our data show that PI3K gamma and delta inhibitors reduced ex vivo NETosis in COVID-19 patients. Alongside other potential benefits including reduced plasma cytokine levels, we suggest that a clinical trial assessing these inhibitors may benefit patients with COVID-19 [57].

We observed an altered neutrophil phenotype in moderate COVID-19, not compatible with previously described populations, and not in keeping with our AMC or CAP control cohorts. COVID-19 neutrophils expressed decreased levels of the activation marker CD11b and a lack of CD62L shedding, which has previously been observed in sepsis [58], alongside reduced levels of PD-L1 involved in immunosuppression. We saw no changes in the expression of CD10; a marker of immature neutrophils [59], or in CXCR2 and CXCR4; markers of senescence [60]. This contrasted with RNAseq studies that reported populations of immature and senescent neutrophils in severe COVID-19 patients compared to mild patients or non-AMC [22,61]. Our contrasting data from moderate COVID-19 patients suggests that either the duration of COVID-19 infection or the extreme severity of infection (from not requiring, to requiring organ support) affects cellular dysfunction and, as in pneumonia and sepsis [17], may support a window for therapeutic intervention.

Finally, we observed that COVID-19 neutrophils expressed elevated CD54, a marker of reverse transmigration, whereby neutrophils migrate from the tissues back into the circulation. These cells are capable of high levels of oxidative burst [62], which may contribute to high levels of NETosis. By day three to five post-admission, we report increased levels of senescent CXCR4^+^ CXCR2^-^ neutrophils, confirming a report of reduced CXCR2^+^ neutrophils in ICU COVID-19 patients [63].

The majority of COVID-19 hospitalisations and deaths occur in non-ICU wards [29], making this cohort important for targeted intervention. While there was evidence of systemic inflammation indicated by elevated levels of circulating IL-6, sTNFRI and VEGF, less than 5% of the patients in the current study met the criteria for the hyper-inflammation phenotype described in ARDS [34] and had levels of circulating mediators lower than described in “usual” sepsis [13]. IL-6 and CXCL-8, as well as platelet derived factors and antigen-antibody complexes, are thought to drive NET formation, providing another mechanism to link systemic inflammation to neutrophil dysfunction [64,65,66]. We did not find significantly elevated levels of GM-CSF in our patients, confirming recent findings in patients of a similar severity [67].

Our data suggests a distinct cellular response in moderate COVID-19 which contributes to on-going immune mediated harm, but which may be modifiable using a targeted therapy, such as PI3Kδ or PI3K γ inhibitors administered at this crucial point in disease progression.

Our data complements studies which showed increased systemic NETosis in COVID-19 [24,68], and elevates those which show increased NETosis in isolated neutrophils, by including increased patient numbers and appropriate AMC [23,24,44]. A study by Masso-Silva et al., recently showed elevated neutrophil phagocytosis in sixteen ICU COVID-19 patients compared to non-AMC. Differing results may be due to the differing experimental techniques, disease severity and patient numbers. Their study used polymorphoprep rather than Percoll^®^ to isolate neutrophils, and the authors combined data from multiple blood samples taken over eleven days of hospitalization. As we show changes in neutrophil phenotype and function over the three to five time course of our study, we suggest that combining time points obscures the complex changes occurring in this short-lived cell population. We also used opsonized *S. pneumoniae* for phagocytosis studies which may be phagocytosed by different mechanisms to *S. aureus* bioparticles [52], confounding results.

More recently, Loyer et al., demonstrated the increased expression of CD11b and ROS production with decreased CD62L expression in a cohort of COVID-19 patients treated in the ICU in comparison with CAP and healthy controls [69]. The authors suggest that this neutrophil dysfunction may be related to neutrophil exhaustion and mortality. This is compatible with our findings in our ward based COVID-19 patients suggesting that the evolution of disease severity can lead to poor patient outcomes through neutrophil exhaustion. Targeting dysfunctional pathways before patients require ICU care could prevent disease progression and death.

### Limitations

This study was limited due to the safety measures required when handling biological fluids for a new infectious disease. All experiments were carried out within a BSL2 hood and methods were chosen based on tolerance to inactivation/fixation with 4% PFA. Our patients did not include an ICU group; however, mild-moderate disease affects a larger proportion of overall COVID-19 patients, and we believe it is this point in the patient pathway which holds most potential for successful intervention. Our AMC group highlights changes in COVID-19 and our CAP group highlights differences between disease types.

## 5. Conclusions

Our study shows that moderate COVID-19 is associated with alterations in neutrophil phenotype, increased migratory capacity and NETosis, and impaired antimicrobial function, which contributes to the severity of COVID-19. Elevated NETosis in the lung is associated with disease severity, and elevated systemic NET production is likely to contribute to inflammation, which may drive ARDS associated damage and thrombosis. Targeting neutrophils and their downstream effectors may be beneficial in the treatment of COVID-19.

## Figures and Tables

**Figure 1 cells-11-02901-f001:**
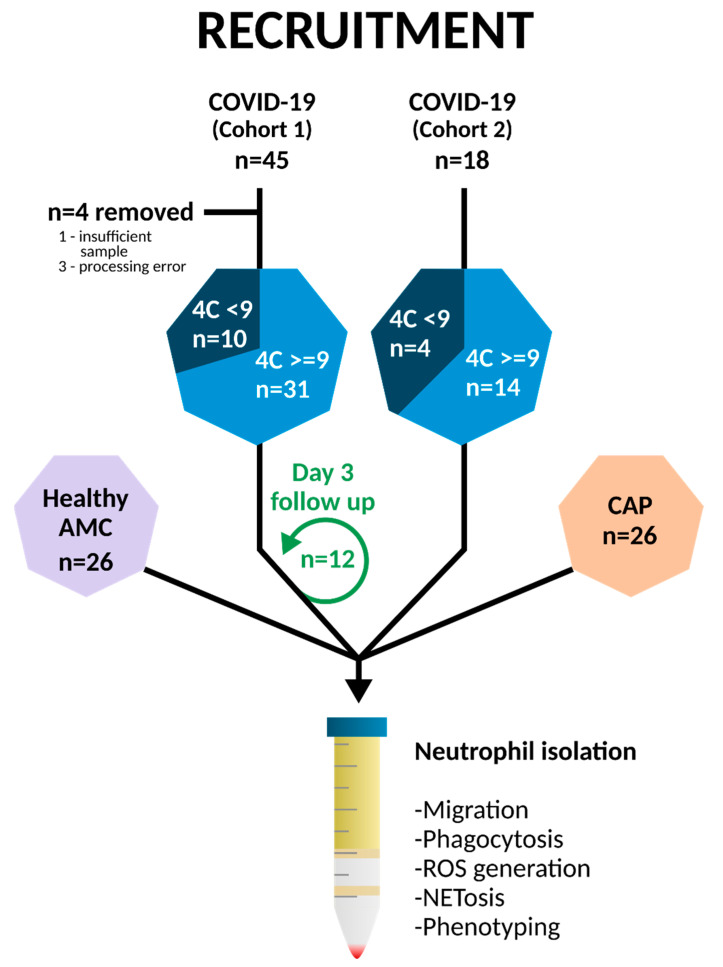
Patient recruitment. 45 hospitalised, non-ICU patients with COVID-19 were recruited from the Queen Elizabeth Hospital, Birmingham from January to March 2021 alongside 26 age matched controls (AMC) and 26 hospitalised patients with non-COVID-19 community-acquired pneumonia (CAP). Four COVID-19 patients were excluded, and 12 patients were re-sampled on days three to five post original sample. A second cohort of 18 COVID-19 patients was recruited between August 2021 and January 2022 for PI3K inhibitor studies. Blood was taken within 48 h of admission, and neutrophils were isolated by Percol density gradient centrifugation. Functional experiments including phagocytosis, NETosis, and phenotype were performed.

**Figure 2 cells-11-02901-f002:**
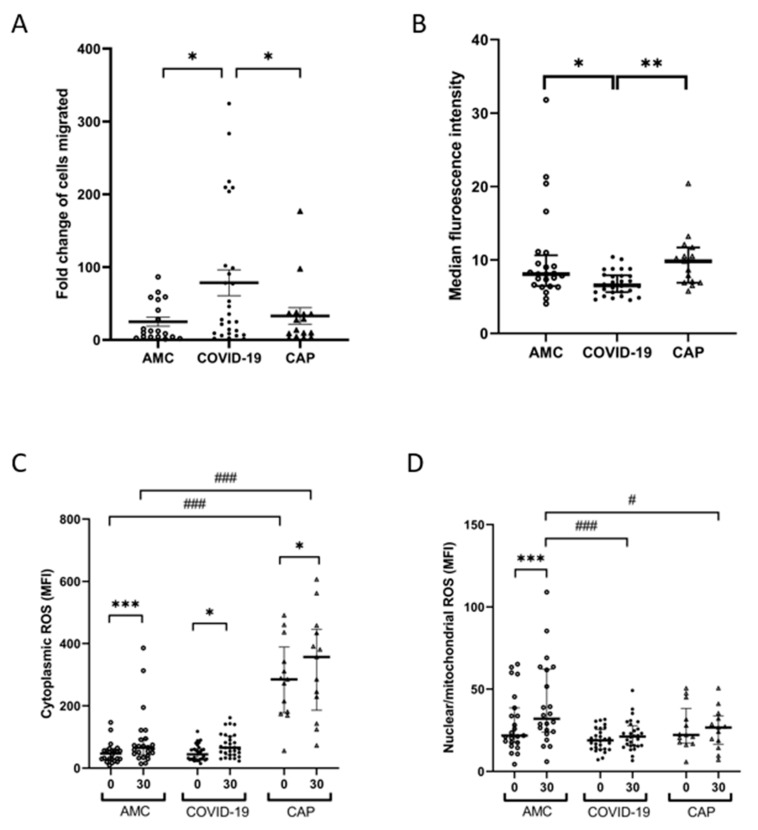
Neutrophil functional responses in COVID-19. (**A**) Neutrophil transmigration towards CXCL-8. Data shows fold change in neutrophils migrated to CXCL-8 compared to vehicle control. * *p* = 0.0332 AMC (n = 18) vs. COVID-19 (n = 28), * *p* = 0.0332 CAP (n = 10) vs. COVID-19 (n = 28). (**B**) Neutrophil phagocytosis of opsonized *S. pneumoniae* following a 30-min incubation. Data shows MFI of positive neutrophils. * *p* = 0.0366 AMC (n = 24) vs. COVID-19 patients (n = 30), ** *p* = 0.0052 CAP (n = 15) vs. COVID-19 (n = 30). (**C**) Cytoplasmic ROS levels measured in neutrophils at baseline (0 min) and after phagocytosis. *** *p* < 0.0091 AMC (n = 24), * *p* = 0.038 CAP (n = 7). CAP patients showed elevated baseline cROS (### *p* < 0.0001) and after phagocytosis (### *p* < 0.0001) vs. AMC. (**D**) Nuclear/mitochondrial ROS levels in neutrophils at baseline (0 min) and after phagocytosis. *** *p* < 0.0001 AMC. COVID-19 (### *p* < 0.0001) and CAP patients (# *p* = 0.0101) showed reduced n/mROS after phagocytosis compared to AMC.

**Figure 3 cells-11-02901-f003:**
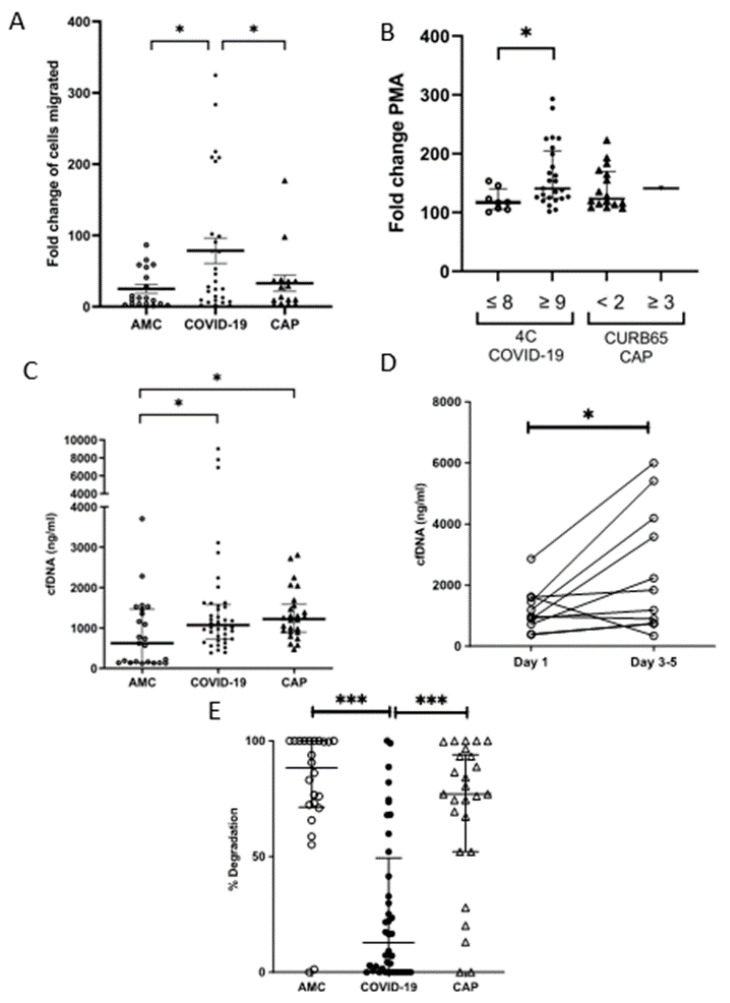
NETosis in COVID-19. (**A**) NET release by neutrophils stimulated with PMA for 3 h, measured as the absorbance of cfDNA stained with Sytox green. Data shows a fold change in absorbance of PMA stimulated neutrophils compared to vehicle control. * *p* = 0.0394 AMC (n = 26) vs. COVID-19 (n = 33), CAP (n = 17). (**B**) Comparison of NET release by activated neutrophils in patients with low 4C (n = 8) or high 4C (n = 26) score, * *p* = 0.0118. There was no difference between CAP patients differentiated by curb65 score. (**C**) Plasma cell free DNA levels measured by fluorometry in AMC * *p* = 0.0396 AMC (n = 23) vs. COVID-19 (n = 39), * *p* = 0.0322 AMC vs. CAP (n = 26). (**D**) Comparison of cfDNA levels measured by fluorometry in COVID-19 (n = 10) on day one and on days three to five. * *p* = 0.0186. (**E**) % DNA degradation by serum NETs. *** *p* < 0.0001 AMC (n = 24) vs. COVID-19 (n = 40), *** *p* = 0.0001 AMC vs. CAP (n = 26).

**Figure 4 cells-11-02901-f004:**
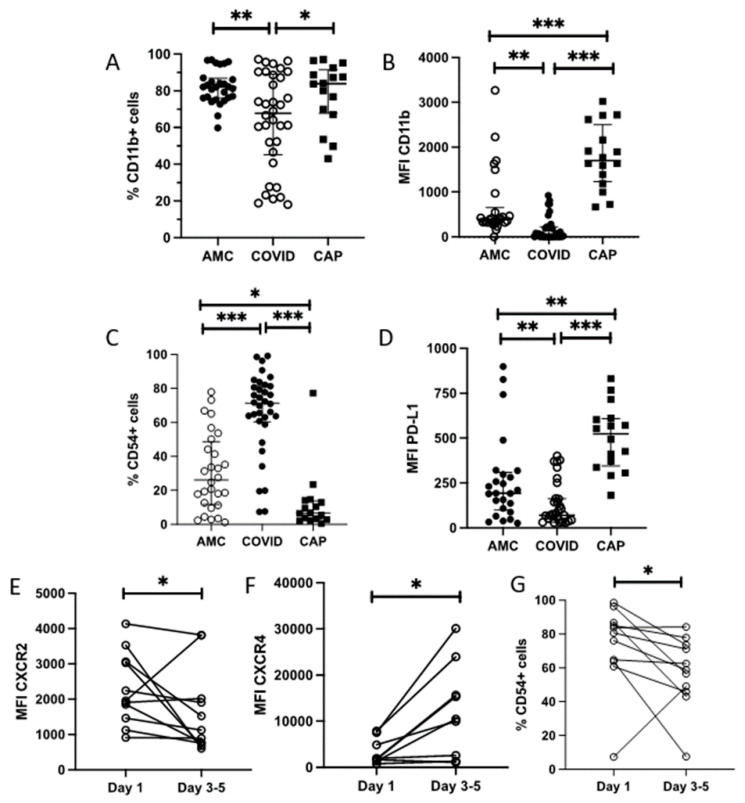
Neutrophil phenotype in COVID-19. Expression of cell surface markers on neutrophils from AMC (n = 26), COVID-19 patients (n = 34) or CAP patients (n = 16) measured by flow cytometry. Data shows % cells expressing CD11b (**A**) *p* = 0.00014, MFI of CD11b (**B**) *p* = 0.0026, % expression of CD54 (**C**) *p* = <0.0001 and MFI of PD-L1 (**D**) *p* = 0.006 in AMC (n = 28), COVID-19 patients (n = 34), CAP patients (n = 16). Data were analysed by one way ANOVA. Expression of CXCR2 (**E**) *p* = 0.0322, CXCR4 (**F**) *p* = 0.0273 and % expression of CD54 (**G**) *p* = 0.0420) on COVID-19 neutrophils on day one or follow up (day three to five). Data are expressed as individual points, analyzed by Wilcoxon test (n = 11). * *p* < 0.05, ** *p* < 0.001, *** *p* < 0.0001.

**Figure 5 cells-11-02901-f005:**
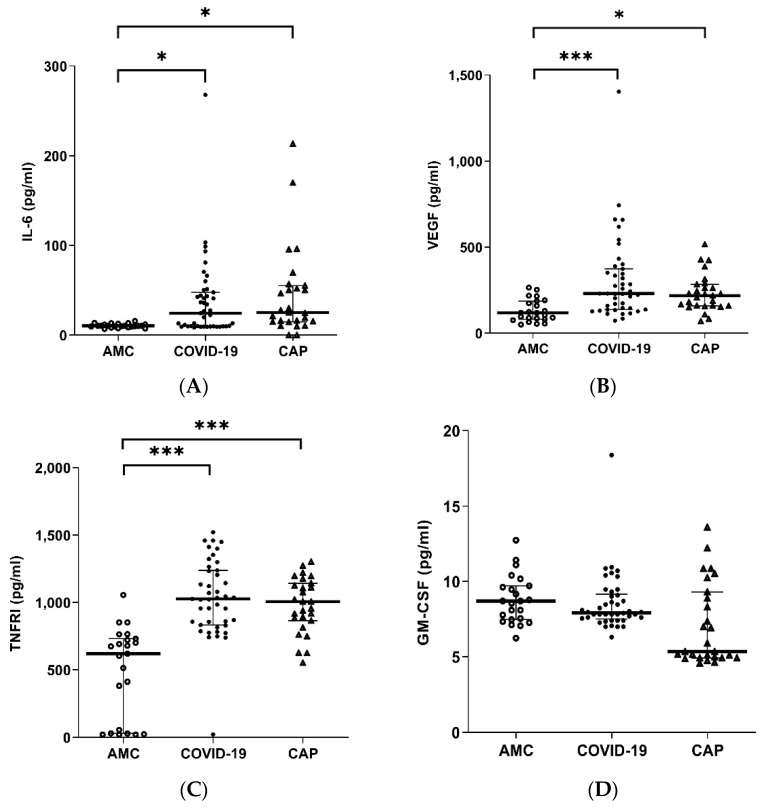
Plasma cytokine levels in COVID-19. Plasma cytokine concentrations in AMC (n = 23), COVID-19 (n = 41), CAP (n = 26) (**A**) IL-6 * *p* = 0.0296 AMC vs. COVID-19, * *p* = 0.0106 AMC vs. CAP. (**B**) VEGF *** *p* < 0.0001 AMC vs. COVID-19, * *p* = 0.0032 AMC vs. CAP. (**C**) sTNFRI *** *p* < 0.0001 AMC vs. COVID-19, *** *p* < 0.0001 AMC vs. CAP. (**D**) GM-CSF overall *p* = 0.3375, (**E**) MPO *** *p* < 0.0001 AMC vs. COVID-19, *** *p* < 0.0001 COVID-19 vs. CAP.

**Figure 6 cells-11-02901-f006:**
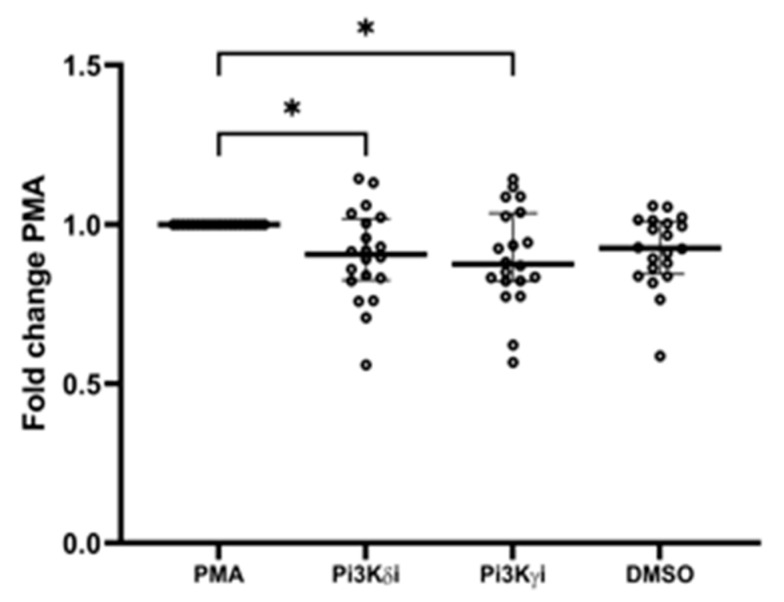
PI3K inhibition. COVID-19 neutrophils were pre-incubated for 30 min with PI3Kσ inhibitor CAL101 and PI3Kγ inhibitor AS252434, and NET release measured in response to PMA stimulation. Data shows fold change in absorbance of PMA stimulated neutrophils compared to PMA control, n = 18. * *p* = 0.0129 PMA vs. PI3KσI, * *p* = 0.0156 PMA vs. PI3KγI.

**Figure 7 cells-11-02901-f007:**
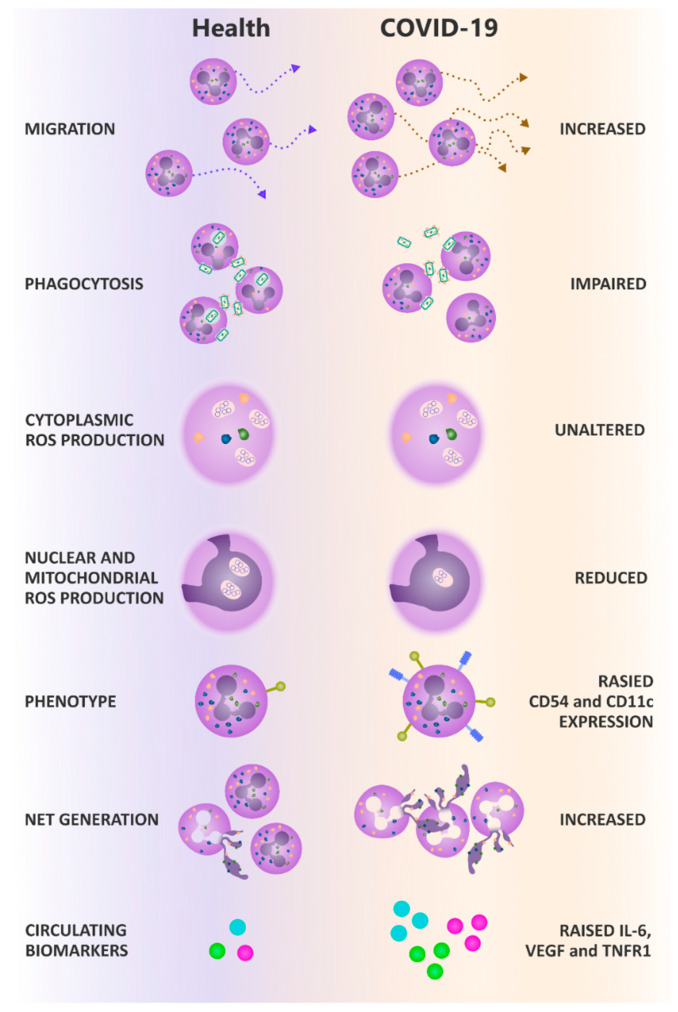
Summary of results. Compared to AMC, neutrophils isolated from non-ICU COVID-19 patients demonstrate increased migration, impaired phagocytosis and reduced nuclear/mitochondrial ROS generation. COVID-19 neutrophils have altered phenotypes, displaying increased expression of migration markers CD54 and CD11b. COVID-19 patients also display elevated NETosis, both ex vivo and in the circulation, and elevated pro-inflammatory cytokines. This contributes to disease pathogenesis in COVID-19.

**Table 1 cells-11-02901-t001:** Recruited patient demographics, collected at the time of enrolment. WCC- white cell count; CRP- C-reactive protein; NLR—neutrophil lymphocyte ratio; NEWS2—National Early Warning Score 2, data collected was worst score in the 24 h after admission; HS Troponin I—high sensitivity troponin I; qSOFA—quick Sepsis-related Organ Failure Assessment Score; CURB-65-Mortality in community acquired pneumonia. Comparisons were calculated between cohorts of COVID-19 patients and AMC, CAP or COVID-19 PI3K patients. * *p* < 0.05, ** *p* < 0.001, *** *p* < 0.0001. Normally distributed result shown with mean (SEM), not normally distributed shown with median (IQR Q1–Q3). Other comorbidities were included if they caused a significant impact on patient quality of life or regular medication—this included but was not limited to severe peripheral vascular disease with ulceration, dementia, chronic kidney disease, stroke, childhood polio, obesity, diverticulosis, alcohol related liver disease, or rheumatoid arthritis.

	AMC n = 26	COVID-19 n = 41	CAP n = 26	COVID-19 Pi3K n = 18
**Male:Female**	10:16	26:15	15:11	9:9
**White:Non-white**	24:2	31:10	23:3	13:5
**Died (%)**	0 (0%)	10 (24%)	2 (8%)	5 (28%)
**Age**	70 (61.0–78.0)	71.5 (58.0–84.0)	67.5 (54.5–85.0)	70.7 (51.0–83.0)
**BMI (kg/m^2^)**		26.4 (24.2–32.8)	27.9 (23.3–41.1)	30.1 (23.2–27.2)
**Comorbidities**				
**Cardiovascular**	4 (15.3%)	12 (29.3%)	7 (26.9%)	7 (38.9%)
**Respiratory**	0 (0%)	1 (2.4%)	4 (15.4%)	2 (11.1%)
**Endocrine**	10 (38.4%)	16 (39.0%)	3 (11.5%)*	9 (50.0%)
**Hypertension**	13 (50%)	19 (46.3%)	10 (38.5%)	9 (50.0%)
**Other**	11 (42.3%)	24 (58.5%)	15 (57.7%)	8 (44.4%)
**WCC (×10^9^/L)**		8.2 (6.3–12.0)	13.4 (10.7–16.4) ***	7.6 (4.6–11.9)
**Neutrophils (×10^9^/L)**		6.4 (4.4–8.6)	11.2 (7.8–13.6) **	5.8 (2.9–9.4)
**CRP (mg/L)**		103.0 (63.0–165.0)	119.0 (42.0–396.0)	74 (24.3–161.3)
**NLR**		5.4 (3.8–10.8)	8.5 (5.9–52.7)	6.3 (2.4–10.2)
**Worst NEWS2**		6.0 (5.0–7.0)	5.0 (3.0–12.0)	6.0 (3.0–8.0)
**HS Troponin I (ng/L)**		14.5 (5.0–31.3)	17.5 (4.0–318.0)	7.0 (4.0–31.5)
**D-dimer (ng/mL)**		382.0 (218.0–829.5)	659.0 (270.5–1510.0)	493.0 (247.0–890.0)
**Ferritin (ug/L)**		1082 (428.3–1525.0)	110.0 (78.8–225.3) *	251.5 (185.8–1230.3)
**Vitamin D (nmol/L)**		35.6 (23.0–51.8)	45.3 (23.8–73.2)	36.6 (22.6–60.1)
**Dexamethasone**		38 (92.6%)	0 (0%)	10 (55.6%) *
**Admission**				
**4C**		12.0 (9.0–14.0)		13 (11.0–15.8)
**qSOFA**		1.0 (1.0–1.5)	1.0 (1.0–1.0)	1.0 (1.0–1.0)
**CURB-65**		2.0 (1.0–3.0)	2.0 (2.0–2.0)	2.0 (2.0–3.0)
**Length of stay (days)**		5.5 (3.0–12.0)	4 (3.0–7.5)	6.5 (3.3–14.8)

**Table 2 cells-11-02901-t002:** Percentage of cells expressing, or MFI of isolated neutrophil cell surface markers measured by flow cytometry in AMC (n = 26), COVID-19 patients (n = 34) or CAP patients (n = 16). Data displayed as median (IQR). Data analyzed by individual one way ANOVA with multiple comparisons. Bold indicates significance where * vs. AMC, # vs. COVID-19 and $ vs. CAP.

Receptors	% Expression				MFI			
	AMC n = 26	COVID n = 34	CAP n = 16	*p* Value	AMC n = 26	COVID n = 34	CAP n = 16	*p* Value
**CD10**	94.3 (11.52)	95.3 (13.6)	74.7 *# (60.6)	* <0.0006 # <0.0006	669 (511)	836 (518)	701 (486)	0.304
**CD11b**	81.6 (15.25)	67.8 *$ (43.9)	83.9 (23.8)	* 0.0014 $ 0.046	399 (331)	40 *$ (223)	1701 * (1268)	* 0.0026 $ <0.0001 * <0.0001
**CD54**	26.2 (37.0)	71.3 *$ (21.2)	6.6 (10.9)	* <0.0001 $ <0.0001	33 (48)	73 (58)	147 *# (77)	* 0.0001 # 0.0012
**CD62L**	23.3 (58.8)	31.0 (15.3)	49.9 (41.4)	0.384	68 (400)	94 (121)	723 *# (519)	* 0.0005 # <0.0001
**CXCR2**	99.9 (0.1)	100 (0.2)	98.5 (1.6)	0.741	2482 (1415)	2031 (1600)	2839 (2839)	0.439
**CXCR4**	93.9 (16.4)	96.7 (9.07)	54.3 *# (35.0)	* <0.0001 # <0.0001	1917 (4039)	1724$ (4177)	1090 (1376)	$ 0.028
**CD66b**	99.7 (0.5)	99.8 (0.3)	97.6 (1.6)	0.298	324 (367)	186 (161)	1631 *# (1493)	* <0.0001 # <0.0001
**CD11c**	99.2 (2.9)	99.9 (1.05)	76.1 *# (64.5)	* <0.0065 # <0.0001	392 (2120)	1148 (670)	4197 *# (6040)	* <0.0001 # <0.0001
**PD-L1**	63.8 (32)	98 (27.9)	8.7 *# (12.9)	* <0.0001 # <0.0001	193 (210)	70 *$ (132)	524 * (264)	* 0.006 $ <0.0001 * 0.0006

## Data Availability

Data can be requested and access will be reviewed on a case-by-case basis by the corresponding author.

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
