# Peer review of "Dysregulated Neutrophil Phenotype and Function in Hospitalised Non-ICU COVID-19 Pneumonia"

_cells, 2022, doi:10.3390/cells11182901_

Round 1

Reviewer 1 Report

The authors studied neutrophil function in hospitalised, non-ICU COVID-19 patients, and compared the results with age matched controls and community acquired pneumonia patients. This is an interesting study, and the results are attractive with observed differences in neutrophil function and phagocytosis. However, the manuscript was not prepared with care. My concerns are listed below: 

Major concerns 

  1. 1. Details of the second cohort of COVID-19 patients were not shown in the main manuscript. 

  1. 2. It was initially stated that there were 23 AMC included in the study, however, in the figure legends: figure 2 line 192, the AMC (n=24), Figure 3 line 235, the AMC (n=26), figure 4 line 281, the AMC (n=28); as well as Table 2 line 249, the AMC (n=28). These should be clarified. 

  1. 3. Supp. Table 1: legend needs to be revised. This table is only about COVID-19 patients, however the legend also described AMC/CAP patients, this is very confusing. P values were given, but none of the parameters were marked as significant.  

Minor concerns: 

  1. 1. In the supplementary figure 6, last sentence, **p<0.0206 should be * p<0.0206, there is not any ** shown on figure D. 

  1. 2. Please explain this sentence: line 94 “patients unable to consent due to lack of capacity were consented by proxy; What proxy means here? 

  1. 3. Figure 1: second COVID-19 cohort should be included  

  1. 4. Figure 2: authors stated cROS levels was not significantly changed after phagocytosis in COVID-19 patients (line 202), however, in the figure (2C) there is a “*” over the COVID-19 patients’ data.  

  1. 5. Lines 101-103: inadequate information in the figure legend, CAP group should be mentioned, and all the abbreviations should be defined. 

  1. 6. In the supplement tables, the COVID-19 variant was not complete.

Author Response

Reviewer 1:

The authors studied neutrophil function in hospitalised, non-ICU COVID-19 patients, and compared the results with age matched controls and community acquired pneumonia patients. This is an interesting study, and the results are attractive with observed differences in neutrophil function and phagocytosis. However, the manuscript was not prepared with care. My concerns are listed below: 

We thank the reviewer for their valued expertise in evaluating our manuscript. We believe the changes made based on your advice have improved our submission.

Major concerns:  

  1. Details of the second cohort of COVID-19 patients were not shown in the main manuscript.

We have edited Table 1 to include this cohort and have thus removed supp table 1 that previously showed this data. All other supplementary table numbers have been updated to reflect this change.

  1. It was initially stated that there were 23 AMC included in the study, however, in the figure legends: figure 2 line 192, the AMC (n=24), Figure 3 line 235, the AMC (n=26), figure 4 line 281, the AMC (n=28); as well as Table 2 line 249, the AMC (n=28). These should be clarified. 

We have double checked all data files and updated Table 1 to show the 26 AMC used in the study. The figure legends have been checked and are correct for each graph – slight changes in n reflect actual numbers used in each experiment. This number changes between experiments as not all samples provided sufficient cells to perform every experiment simultaneously.

  1. Supp. Table 1: legend needs to be revised. This table is only about COVID-19 patients, however the legend also described AMC/CAP patients, this is very confusing. P values were given, but none of the parameters were marked as significant.

This table has now been removed and table 1 updated as above with statistics re-performed.

Minor concerns: 

  1. In the supplementary figure 6, last sentence, **p<0.0206 should be * p<0.0206, there is not any ** shown on figure D.

This has been changed as requested

  1. Please explain this sentence: line 94 “patients unable to consent due to lack of capacity were consented by proxy;” What proxy means here?

We have added a line in supplementary text under ‘patients’ to explain this.

‘Patients unable to consent due to lack of capacity were consented by ‘proxy’ which was either next of kin or consultant in charge of care’.

  1. Figure 1: second COVID-19 cohort should be included 

We have changed figure 1 as requested

  1. Figure 2: authors stated cROS levels was not significantly changed after phagocytosis in COVID-19 patients (line 202), however, in the figure (2C) there is a “*” over the COVID-19 patients’ data.

We thank the reviewer for highlighting this error. There should not be a * over the COVID data – this has been removed and the correct graph included.

  1. fIGURE 1 Lines 101-103: inadequate information in the figure legend, CAP group should be mentioned, and all the abbreviations should be defined.

We have re-written the legend as below to correct this error:

45 hospitalized, non-ICU patients with COVID-19 were recruited from the Queen Elizabeth Hospital Birmingham from January to March 2021, alongside 28 age matched controls (AMC) and 26 hospitalized patients with non-COVID-19 community-acquired pneumonia (CAP). 4 COVID-19 patients were excluded, and 12 patients were re-sampled on days 3-5 post original sample. Blood was taken within 48 hours of admission, and neutrophils isolated by percol density gradient centrifugation. Functional experiments including phagocytosis, NETosis, and phenotype were performed.

  1. In the supplement tables, the COVID-19 variant was not complete.

We have removed this heading from the table as unfortunately we were unable to obtain complete variant data for patients as we had hoped.

Reviewer 2 Report

In this manuscript, by comparing neutrophil phenotype and function in covid-19 patients, AMC, and CAP patients, the author found that covid-19 patient-derived neutrophils altered phenotype, elevated migration and NETosis, and impaired antimicrobial responses. In general, the experiment design was proper and the data is clear to support the conclusion. However, I still have some concerns about this work.

1. Can NETs be measured in neutrophils direct after isolation? Why do the authors add additional PMA to stimulate neutrophils? PMA is a strong activator for NETs formation in neutrophils, can this experiment reflect the real physiological condition in vivo?

2. I am quite confused with the multiple parameters the author analyzed in neutrophils, like ROS production, migration, NETs formation, and phagocytosis. Each of these parameters is critical to neutrophil functions. The authors observed decreased ROS production, generally, which may mean less inflammation and less tissue damage, however, the migration and NETs were elevated, which means the neutrophils were more aggressive, which may cause more tissue damage. Although the authors claim that the phenotype of covid-derived neutrophils was different from AMC and CAP-derived neutrophils and they may contribute at different disease stages, Do they counterbalance? which parameter the author examined may be tightly related to covid-19 progression? or which parameter could be the key driver for covid-19 progression?

3. cROS and n/m ROS, which one is important for neutrophil activation in covid? How to explain only n/m ROS decreased in covid-19-derived neutrophils? n/m ROS had no significant increase 30 minutes after phagocytosis in covid-derived neutrophils, which may mean n/m ROS didn't respond to phagocytosis at all. 

Author Response

Reviewer 2

In this manuscript, by comparing neutrophil phenotype and function in covid-19 patients, AMC, and CAP patients, the author found that covid-19 patient-derived neutrophils altered phenotype, elevated migration and NETosis, and impaired antimicrobial responses. In general, the experiment design was proper and the data is clear to support the conclusion. However, I still have some concerns about this work.

We thank the reviewer for taking time to evaluate our manuscript and hope you will agreed the changes have improved our manuscript.

  1. Can NETs be measured in neutrophils direct after isolation? Why do the authors add additional PMA to stimulate neutrophils? PMA is a strong activator for NETs formation in neutrophils, can this experiment reflect the real physiological condition in vivo?

Basline NETosis data was obtained but was not significantly altered. As such this data wasplaced in supplementary data (figure 6D). PMA is added to neutrophils in order to stimulate a maximal NETosis response through the canonical pathway. Unstimulated neutrophils release small levels of NETs upon isolation, and so we did not expect to see any differences without stimulation – as shown in figure supplementary 6D which was not significantly different. We have chosen to display the PMA-stimulated data as this is more likely to reflect the ability of neutrophils to release NETs when they reach the lungs and encounter the highly inflammatory environment induced by COVID-19.

  1. I am quite confused with the multiple parameters the author analyzed in neutrophils, like ROS production, migration, NETs formation, and phagocytosis. Each of these parameters is critical to neutrophil functions. The authors observed decreased ROS production, generally, which may mean less inflammation and less tissue damage, however, the migration and NETs were elevated, which means the neutrophils were more aggressive, which may cause more tissue damage. Although the authors claim that the phenotype of covid-derived neutrophils was different from AMC and CAP-derived neutrophils and they may contribute at different disease stages, Do they counterbalance? which parameter the author examined may be tightly related to covid-19 progression? or which parameter could be the key driver for covid-19 progression?

Reviewer 2 is correct in that each of these parameters is critical for neutrophil function, which is why we took a holistic approach to measure all in the same patient samples. Decreased ROS production is not unexpected in this study, as it may be a mechanism linked to decreased bacterial killing. However, we were unable to directly measure bacterial killing in this study, due to the need to fix cells prior to analysis. However, both the ROS data and the phagocytosis show that the antimicrobial response of COVID-19 neutrophils is impaired. This may contribute to the increase in secondary infection seen in COVID patients, which is the leading cause of death in this disease. Increased migration and NETs may act co-currently to contribute to tissue damage in the lungs. We believe these responses are linked, but not necessarily counterbalancing each other. We believe that NET production contributes to tissue damage, as suggested by other authors in this topic, but believe it would be an overreach to conclude that NETosis is directly related to disease progression based on our data alone.

  1. cROS and n/m ROS, which one is important for neutrophil activation in covid? How to explain only n/m ROS decreased in covid-19-derived neutrophils? n/m ROS had no significant increase 30 minutes after phagocytosis in covid-derived neutrophils, which may mean n/m ROS didn't respond to phagocytosis at all. 

N/M ROS was significantly reduced in COVID-19 neutrophils compared to AMC (FIG 2D) but was not elevated after phagocytosis – however it is elevated in AMC. This may mean there is a defect in COVID-19 patient neutrophils, leading to impaired ability to increase n/mROS, linked to impaired antimicrobial capacity. We refer to this in line 346-348. ROS generation is not linked to neutrophil activation but is a response to the phagocytosis of bacteria. We believe that the inability of covid neutrophils to generate an increase in n/mROS may be linked to impaired bacterial killing, but were unable to directly measure this in our study due to limitations requiring inactivation of samples before analysis. It would be interesting to see how long this change in ROS and bacterial phagocytosis persisted over a longer time course, but this was not possible in this study.

Round 2

Reviewer 2 Report

The authors addressed most questions and the paper is good to publish.